# Ultrafiltration Membranes Functionalized with Copper Oxide and Zwitterions for Fouling Resistance

**DOI:** 10.3390/membranes12050544

**Published:** 2022-05-23

**Authors:** Cannon Hackett, Mojtaba Abolhassani, Lauren F. Greenlee, Audie K. Thompson

**Affiliations:** 1Ralph E. Martin Department of Chemical Engineering, University of Arkansas, Fayetteville, AR 72701, USA; cjhacket@uark.edu (C.H.); abolhassani.mo@gmail.com (M.A.); 2Department of Chemical Engineering, Pennsylvania State University, University Park, PA 16802, USA; lfg5271@psu.edu

**Keywords:** anti-fouling, ultrafiltration, polydopamine, copper oxide, zwitterion

## Abstract

Polymeric membrane fouling is a long-standing challenge for water filtration. Metal/metal oxide nanoparticle functionalization of the membrane surface can impart anti-fouling properties through the reactivity of the metal species and the generation of radical species. Copper oxide nanoparticles (CuO NPs) are effective at reducing organic fouling when used in conjunction with hydrogen peroxide, but leaching of copper ions from the membrane has been observed, which can hinder the longevity of the CuO NP activity at the membrane surface. Zwitterions can reduce organic fouling and stabilize NP attachment, suggesting a potential opportunity to combine the two functionalizations. Here, we coated polyethersulfone (PES) ultrafiltration membranes with polydopamine (PDA) and attached the zwitterionic compound, thiolated 2-methacryloyloxyethyl phosphorylcholine (MPC-SH), and CuO NPs. Functionalized membranes resulted in a higher flux recovery ratio (0.694) than the unfunctionalized PES control (0.599). Copper retention was high (>96%) for functionalized membranes. The results indicate that CuO NPs and MPC-SH can reduce organic fouling with only limited copper leaching.

## 1. Introduction

Ultrafiltration membranes are used in various applications, including water purification, protein separation, and food processing [1]. Although ultrafiltration membranes have become increasingly used over the past several decades, fouling has continued to pose a major challenge in their implementation [1]. Fouling hinders the efficiency and longevity of ultrafiltration membranes by reducing the flux through the membranes. This additional resistance to permeation leads to higher energy requirements or cleaning requirements. As a result, membranes with unique features can be designed. In particular, surface alterations are aimed at controlling the separation of specific feed sources (i.e., wastewater, produced water, etc.), while both polymer matrix and surface modifications can simultaneously improve flux performance and impart antifouling properties [2]. There are four main types of fouling: organic, inorganic, colloidal, and biological [3]. Various strategies have been pursued to mitigate and counteract fouling [4], but these also have drawbacks. For instance, membranes can be cleaned using strongly alkali, acidic, or chlorinated chemical solutions to remove foulants, but these solutions can damage the membranes [5].

As an alternative, membranes can be functionalized with reactive nanoparticles to improve resistance to fouling. Nanoparticles are useful in the context of membranes due to their unique properties, as compared to the bulk material, and possible synergy between the nanoparticles and polymer in the membrane [6]. While most research thus far in this arena has focused on metallic Ag NPs and surface attachment [7,8,9], several studies have demonstrated successful incorporation of oxide NPs through zwitterion attachment and incorporation into the membrane polymer matrix during membrane formation [10,11]. Nanoparticles (NPs) made from materials including Cu [12], CuO [13,14,15], Ag [16], Al_2_O_3_ [17], SiO_2_ [18], ZnO [19], and TiO_2_ [20], have been shown to have antifouling effects against various organic and inorganic foulants and bacteria. Amongst the different NP compositions, metallic NPs suffer from high metal leaching rates because the mechanism of reactivity is through oxidation and dissolution of the metal cation (i.e., Ag^0^ NPs release Ag^+^) [21,22]. Some NPs such as ZnO and TiO_2_ can agglomerate if added to the polymer dope solution [23]. Additionally, some oxides are minimally reactive (e.g., SiO_2_, Al_2_O_3_) or require ultraviolet light exposure (TiO_2_) to impart anti-fouling reactivity. In contrast, CuO NPs are of particular interest because the oxide phase is more stable in aqueous environments than metallic NPs and the CuO composition can react with additives such as hydrogen peroxide to effectively clean membrane surfaces without destroying the polymer. In addition, CuO NPs have benefits including low cost, low toxicity, and customizability to various shapes and sizes [14].

Guha et al. [13] fabricated reverse osmosis membranes coated with CuO NPs and tested the effect of dosing the membranes with hydrogen peroxide. The CuO NPs present on the membrane catalytically decomposed hydrogen peroxide, forming oxygen bubbles that cleared away foulants (silica and humic acid) from the membrane surface and allowed flux to recover. Arumugham et al. [14] fabricated ultrafiltration membranes blended with CuO NPs and graphitic carbon nitride and found that the additives improved resistance to bovine serum albumin (BSA) fouling. The mechanism improved hydrophilicity and surface smoothness, as well as a negative charge, contributed by CuO, which repelled BSA. Krishnamurthy et al. [24] found that PES ultrafiltration membranes incorporated with Cu_2_O NPs reduced fouling by BSA, humic acid, and oil. Zareei et al. [15] determined that PES nanofiltration membranes incorporated with CuO/CoFe_2_O_4_ NPs had improved protein fouling resistance and metal ion rejection. Moreover, PES membranes are commonly used in water filtration and are known for maintaining physical properties under a wide temperature range [13,25].

There are two primary methods for fabricating membranes with CuO NPs; (1) the NPs can be blended into the polymer solution before a membrane is cast [26], or (2) the NPs can be coated onto the surface of a membrane. When the coating strategy is used, membranes are often first coated with polydopamine (PDA), which provides a surface on which to anchor the CuO NPs [13]. However, even with PDA as an anchor for CuO NPs, NP leaching remains a challenge for repeat membrane use and long-term filtration. For instance, Guha et al. [13] detected 70 ppb of copper in the concentrate in their study using CuO NPs attached to membranes by PDA. Ben-Sasson et al. [12] attached Cu and CuO NPs to membranes and found that 30% of the loaded copper dissolved within two days. Zhao et al. [27] determined that for ultrafiltration membranes blended with Cu_x_O, the rate of copper leaching was heavily dependent on the pH of the filtrate solution, as far more copper was leached at pH = 4.0 compared to pH = 7.5. Saraswathi et al. [28] modified polyvinylidene fluoride (PVDF) ultrafiltration membranes by attaching polyhexanide-coated copper oxide nanoparticles, and found that the concentration of nanoparticles in all interval-based samples during the membrane separation was remarkedly low. These results suggest that involving a secondary additive (e.g., polymer, ligand, or zwitterion) would improve CuO NP binding to the membrane surface without compromising filtration. The selection of a zwitterion-type coating, as presented in this paper, would serve a dual purpose of both NP attachment stability and anti-fouling properties.

Interestingly, zwitterionic compounds have also been used to reduce fouling on membranes. These compounds increase the hydrophilicity of the membrane by keeping a layer of water held close to the surface, thus reducing the ability of foulants to adhere to the surface [29,30]. Thiol-based binding of zwitterions has been widely demonstrated for gold surfaces [31], however, the same thiol-based S-metal binding chemistry has also been used to functionalize the surfaces of other metallic compositions [32]. Few studies have explored CuO NP functionalization with zwitterions, but Qasem et al. [33] recently reported thiol attachment to CuO NPs as part of sensor technology. Bengani-Lutz et al. [34] tested four zwitterionic copolymers and found that the zwitterion 2-methacryloyloxyethyl phosphorylcholine (MPC) achieved the highest hydrophilicity and best anti-fouling performance against a protein solution. Thiolated 2-methacryloyloxyethyl phosphorylcholine (MPC-SH), first used to coat biosensors [35], has been used to coat reverse osmosis (RO) membranes, resulting in reduced protein adsorption and bacterial adhesion [36]. Previous work also suggests that MPC-SH could be attached to CuO NPs by the thiol group on MPC-SH since, at room temperature, 1-dodecanethiol binds to CuO, forming a Cu-S bond and reducing the Cu^2+^ to a Cu^+^ oxidation state [37]. However, the ability of MPC-SH to bind to CuO NPs, enhance anti-fouling behavior, and prevent Cu leaching during membrane filtration remains unexplored, but the coupled approach of MPC-SH and CuO NP functionalization of the membrane surface could provide a more robust anti-fouling strategy.

Therefore, this study aimed to modify PES ultrafiltration membranes with a novel combined CuO/MPC-SH functionalization, test the effect of the functionalization on membrane properties and performance, and explore the nature of the bond between CuO NPs and MPC-SH. The successful development of this fabrication strategy can be extended to other nanoparticle materials, zwitterion structures, attachment strategies, and membrane surface and pore modification approaches. The results impact research fields in biomaterials design, antimicrobial coatings and functional materials, biomedical materials, biopharma filtration, and across the research space of water and wastewater filtration [38]. CuO NPs were synthesized and characterized by transmission electron microscopy (TEM) and X-ray diffraction (XRD), while MPC-SH was characterized by mass spectrometry. PDA was used to anchor the MPC-SH to the membranes, and CuO NPs were subsequently attached. The chemistry of CuO NP attachment was investigated through X-ray photoelectron spectroscopy (XPS). The effect of the MPC-SH/CuO NP functionalization on resistance to membrane fouling was evaluated through dead-end filtration with BSA. Finally, the effect of H_2_O_2_ treatment on flux recovery was investigated. The leaching of Cu was assessed during filtration studies with inductively coupled plasma mass spectrometry (ICP-MS) measurements of filtrate samples.

## 2. Materials and Methods

### 2.1. Materials

Commercial PES ultrafiltration membranes with a 50 kDa molecular weight cutoff (model: MQ MAX, 47 mm in diameter) were acquired from Synder Filtration. 2-Methacryloyloxyethyl phosphorylcholine (MPC), acetone, chloroform (anhydrous), dopamine hydrochloride, hydrogen peroxide (3%), sodium hydroxide, trimethylamine (25% in water), and Trizma base were acquired from Sigma-Aldrich (St. Louis, MO, USA). 1,10-decanedithiol was acquired from Alfa Aesar (Haverhill, MA, USA). Diisopropylamine was acquired from Merck (Kenilworth, NJ, USA). Nitric acid (68%) was acquired from VWR (Radnor, PA, USA). Bovine serum albumin (BSA) was acquired from Bio Basic (Markham, ON, Canada). Copper standard (1000 ppm in 5% nitric acid) was acquired from Acros Organics (Waltham, MA, USA). Polyvinylpyrrolidone (PVP) with 40,000 MW was acquired from TCI (Tokyo, Japan).

### 2.2. MPC-SH Synthesis

Thiolated-2-methacryloyloxyethyl phosphorylcholine (MPC-SH) was synthesized by combining 1.48 g of MPC, 2.10 g of 1,10-decanedithiol, and 27.9 µL of diisopropylamine dissolved in 20 mL chloroform (Figure 1). The reaction was flushed with nitrogen gas, then capped. After running the reaction under stirring for 8 h or 20 h at room temperature, 50 mL of acetone was added, which caused a white precipitate to form. The reaction was again flushed with nitrogen gas and capped for 3 days, which allowed the product to settle into a residue at the bottom. The chloroform/acetone liquid was poured off and the residue was dried for three hours using a vacuum desiccator, dissolved in water, then freeze-dried until a stringy white powder was formed. A small sample of the MPC-SH product was taken and diluted in DI water to 0.1 mg/mL. Mass spectroscopy was performed to confirm that MPC-SH was present.

### 2.3. Copper Oxide Nanoparticle Synthesis

The CuO nanoparticles were synthesized by a modified method using a CEM Discover reactor. To synthesize CuO nanoparticles, stock solutions of copper acetate (0.2 M), PVP (0.05 M) and ethylene glycol (with 25 mM sodium hydroxide) were prepared in advance. Then 5 mL of ethylene glycol, 1.5 mL of PVP and 1 mL of copper acetate solutions were mixed using a vortex for 30 s, followed by sonication for 3 min. The sealed borosilicate tube with the mixture was placed in the reactor. The microwave was programmed to ramp the temperature to 110 °C in 7 min and with a holding time of 3 min at 110 °C with power set to 80 W. After the reaction, the vessel was cooled to room temperature (23 °C). The color of the mixture changed to black from light blue. The final nanoparticles were then transferred into the centrifuge tube and washed twice using DI water. The final yield of CuO NPs per batch was measured to be 14.5 mg.

Additionally, CuO/MPC NPs were prepared for XPS analysis to compare with functionalized membranes and to determine the nature of bonds that can be formed between CuO NPs and MPC-SH. CuO/MPC NPs were fabricated by preparing 2 mL of a solution of 10 µL/mL trimethylamine with 22 mg MPC-SH and 5.8 mg CuO NPs. After 16 h, samples were prepared for XPS by applying a drop of the solution to an aluminum substrate. The samples were airdried and analyzed after four days of storage under air.

### 2.4. Membrane Functionalization

Membranes were functionalized with PDA, MPC-SH, and CuO NPs in a three-step process. Prior to functionalization, PES ultrafiltration membranes were rinsed with DI water to remove preservatives. First, membranes were coated with PDA by floating the membranes face-down in beakers containing 10 mL of a solution of 2 mg/mL dopamine HCl and 10 mM Trizma in DI water. After 3 h, the PES/PDA membranes were removed and rinsed with DI water using a wash bottle, then rinsed by soaking in 100 mL DI water for 1 h. Next, MPC-SH was attached to the polydopamine-coated PES membranes by floating the membranes face-down in beakers with 10 mL of a solution of 10 µL/mL trimethylamine and 3 mg/mL MPC-SH in DI water. Membranes were kept in the solution for 16 h, and then the PES/PDA/MPC membranes were rinsed with DI water and soaked in 100 mL DI water for 1 h. Finally, 14.5 mg of CuO NPs (one batch) were sonicated in 10 mL of DI water. Membranes were submerged in the CuO NP suspension for 6 h to finish the fabrication of the PES/PDA/MPC/CuO membranes, then rinsed with DI water and soaked in 100 mL DI water for 1 h to clean off loosely attached nanoparticles.

CuO NPs were hypothesized to attach to the membrane surface via two possible mechanisms. One possibility is that the copper in the CuO NPs would covalently bond to the sulfur atoms in MPC-SH (Figure 2A). An alternative possibility is that the CuO NPs would adsorb on top of the MPC-SH layer (Figure 2B).

For leaching tests, PES/PDA/CuO membranes were also prepared using the method described above, only without the MPC-SH attachment step. In this case, CuO NPs would attach by to PDA by physisorption.

### 2.5. Nanoparticle and Membrane Characterization

To confirm the CuO synthesis high-resolution x-ray diffraction (XRD, Philips X’Pert- MRD diffractometer, Cu Kα radiation source, Amsterdam, Netherlands) was used. XRD patterns were taken within the recorded region of 2θ from 10 to 80° at a voltage of 45.0 kV and a current of 40.0 mA with a scanning speed of 1 min^−1^.

Transmission electron microscopy (TEM, JEOL JEM-1011, Tokyo, Japan) was used to image the CuO NPs and determine their size and shape. A drop of CuO NP suspension was diluted with water, and then a TEM grid was submerged in the suspension and dried. Images were taken with at 500,000× magnification.

Scanning electron microscopy (SEM) was used to examine the effect of the functionalization on the membrane surface and to confirm the attachment of CuO NPs. Membrane samples at each stage of functionalization were airdried overnight and then sputter-coated with platinum for 3 min to reduce charge accumulation before being imaged with the SEM (Philips XL30 Environmental Scanning Electron Microscope, Amsterdam, Netherlands). Additionally, energy-dispersive X-ray spectroscopy (EDX) was used to measure the concentration of elements, including copper, on the surface of the membrane.

To confirm the presence of CuO on the membranes and to determine the nature of the bond between the CuO and the membrane, samples of the functionalized membranes were airdried overnight and analyzed using X-ray photoelectron spectroscopy (XPS, Perkin Elmer PHI 5600ci ESCA System, Waltham, MA, USA)). CuO NPs and CuO/MPC NPs were also analyzed with XPS. Spectrums were calibrated by setting the adventitious carbon peak to 285 eV.

The infrared (IR) spectra of membrane samples were measured with a Fourier transform infrared spectrometer (FTIR, PerkinElmer, Waltham, MA, USA) to confirm the attachment of MPC-SH. Before taking spectra, the PES membrane was rinsed with water to remove preservatives, and all samples of membranes were dried overnight. Finally, the contact angle between membranes and a water droplet was measured using a contact angle instrument (Future Digital Scientific, model OCA15EC, Westbury, NY, USA) with a 5 µL droplet, 0.5 µL/s deposition rate [39]. The contact angle was measured 10 s after the droplet was deposited.

### 2.6. Membrane Filtration

Dead-end filtration experiments were carried out using a Sterlitech HP4750 stirred cell with an effective area of 13.4 cm^2^. The cell was pressurized using compressed nitrogen. The permeate was collected into a container on a balance, which measured the change in permeate mass over time [27,40].

Four types of tests were performed in sequence on each membrane: pure water flux, BSA flux, recovery flux before H_2_O_2_ treatment, and recovery flux after H_2_O_2_ treatment. Each test was performed with 200 mL of water (or BSA solution) in the filtration cell and 35 psi pressure. In each test, flux was measured for 50 min: the membrane was allowed to compact and reach equilibrium flux during the first 30 min and the final flux was calculated based on the last 20 min of data. The BSA flux test used 2 mg/mL BSA in DI water. Before the recovery flux test, the membrane was rinsed by running dead-end filtration with DI water in the inverted position for 1 min. After the recovery flux test, the membrane was rinsed in 20 mL of 3% hydrogen peroxide for 20 min to clean the membrane. Then the membrane was rinsed with DI water, and another dead-end filtration test was performed, again using DI water. Experiments were repeated in triplicates. Flux recovery ratio (FRR) was calculated by:FRR (%) = J_wr_/J_w0_ × 100%(1)
where J_wr_ is the pure water flux of the rinsed membrane and J_w0_ is the starting pure water flux.

### 2.7. Copper Leaching

To determine whether MPC-SH can reduce the level of copper leaching, dead-end filtration tests were run on freshly prepared PES/PDA/CuO membranes and PES/PDA/MPC/CuO membranes. For each membrane, dead-end filtration was performed using 200 mL DI water under 35 psi pressure with 300 rpm stirring. In each case, the effluent was collected, and the copper concentration was measured by inductively coupled mass spectrometry (ICP-MS). After dead-end filtration, the membranes were submerged in 10 mL 2% nitric acid overnight to dissolve the CuO NPs. The concentration of copper was tested using ICP-MS in order to determine the amount of copper that had remained on the membrane and was not leached out during filtration. The fraction of copper retained was calculated by:Retention = (V_m_C_m_)/(V_m_C_m_ + V_p_C_p_)(2)
where V_m_ is the volume of the nitric acid solution in which the membrane had been soaked after filtration, C_m_ is the concentration of copper in that solution, V_p_ is the volume of permeate, and C_p_ is the concentration of copper in the permeate.

## 3. Results and Discussion

### 3.1. MPC-SH Synthesis

Mass spectroscopy was performed on the synthesized MPC-SH. The desired product, MPC-SH, was detected at the *m*/*z* 502.2 peak (spectrum shown in Appendix A). A smaller amount of a side product was detected at the *m*/*z* 797.4 peak. This side-product occurs when MPC attaches on both ends of 1,10-decanedithiol instead of only one end (Appendix A). Therefore, the side product does not have any free thiol groups that are available to react and attach to PDA. Using a reaction time of 8 h led to reduced side product formation and relatively higher peaks for MPC-SH, compared to a reaction time of 20 h (Appendix A), the duration used by Zhao et al. [36]. The 8-h reaction time was subsequently used for this study to minimize the presence of side product in the MPC-SH zwitterion sample.

### 3.2. Nanoparticle and Membrane Characterization

XRD was performed on the synthesized nanoparticles (Figure 3). Peaks associated with CuO were observed at 2θ = 35.2°, 38.5° 48.2°, 53.9°, 61.5°, 66.6°, and 74.6°, in agreement with those reported by Padil et al. [41] for monoclinic, single-phase CuO nanoparticles (JCPDS-05-0661). The broad peaks are indicative of small nanoparticles [42] as opposed to large particulate or bulk CuO [43]. Interestingly Padil et al. [41] also report a peak measured at ~23°, similar to that observed in our synthesized nanoparticles, although other studies in previous literature do not observe a similar peak. However, multiple studies on the diffraction patterns of PVP (e.g., PVP-nanoparticle composites; PVP films) suggest that this broad peak at 23° could likely result from the PVP used during the synthesis procedure [44,45,46,47,48,49]. Generally, a broad peak in the range of 23–25° is indicative of amorphous carbon [46,50]. PVP is present in the synthesis solution as a stabilizer to control CuO nanoparticle size, and a portion of the PVP likely remains with the sample. In addition, Ambalagi et al. [51] report on the synthesis of CuO nanoparticles stabilized by polyaniline and show XRD spectra with a similar broad peak at ~25°. It is also possible that the broad peak at 23° results from amorphous silicates dissolved from the borosilicate glass tube, as sodium hydroxide and other bases are known to cause slow leaching of silica from glassware.

The NPs were approximately 6–10 nm in size (Figure 4). Some particles were nearly spherical, while others were irregularly shaped. Some agglomeration occurred, although the nanoparticles can still be clearly resolved. It is not clear whether the nanoparticles were agglomerated in suspension or if the agglomeration occurred due to drying the sample on the TEM grid.

XPS was performed on synthesized CuO NPs, CuO/MPC NPs, and on the top surface of a PES/PDA/MPC/CuO membrane (Figure 5). For CuO NPs (Figure 5a), in the Cu 2p region, a Cu2p_1/2_ peak was observed at 953.2 eV and a Cu2p_3/2_ peak at 933.8 eV and a shakeup from ~945–940 eV. The 933.8 eV peak is consistent with the literature value of 933.6 eV for CuO [52]. In the O 1s region, there is an O 1s peak at 529.8 eV (S2) from the O^2−^ bonded to copper in CuO [53]. The O 1s peak at 531.6 eV (S1) is extra-lattice oxygen [53], which indicates the non-stoichiometric nature of the surface of the CuO NPs.

For CuO/MPC NPs (Figure 5b), peaks in the Cu 2p region are shifted to lower binding energies than the peaks in the unfunctionalized CuO NPs. The Cu 2p_3/2_ peak at 932.6 eV (shifted down from 933.8 eV for plain CuO NPs) is close to the literature values of 932.5 eV for Cu_2_S and 932.2 eV for CuS [52], which indicates that a bond was formed between the copper in the CuO NPs and the thiol group in the MPC-SH. Similarly, Wang et al. [37] observed shifts in the Cu 2p_3/2_ peak position from 933.8 eV to 931.9 eV upon reaction of CuO NPs with 1-dodecanethiol. An O 1s peak at 532.4 eV (S3) corresponds to organic C-O bonds and the O 1s peak at 530.8 eV (S4) corresponds to C=O bonds, which arise from the MPC-SH. An N 1s peak is visible at 402.4 eV, arising from the alkyl ammonium group in MPC-SH. In the S 2p region, peaks for several sulfur species are present. An S 2p_1/2_ peak is visible at 164.4 eV (S6) and an S 2p_3/2_ peak is visible at 163.4 eV (S7), which are attributed to sulfur in the C-S-C bond formation in MPC-SH [54]. S 2p peaks are present for oxidized sulfur at 168.0 eV (S5) and copper-bonded sulfur at 161.8 eV (S8). The peak at 161.8 eV (S8) is consistent with the literature value for copper sulfide [37] and gives further confirmation of bonding between the thiol group in MPC-SH and the copper in CuO NPs.

For the fully functionalized PES/PDA/MPC/CuO membrane (Figure 5c), in the Cu 2p region, peaks were present at 953.2 eV and 933.8 eV, which are nearly identical positions to the peaks for unfunctionalized CuO NPs. Similarly, the peaks in the O 1s region, at 531.8 eV (S9) and 529.8 eV (S10), are at nearly identical positions the peaks in the O 1s region for CuO NPs. Since the peak positions are essentially unchanged compared to plain CuO NPs, there is no evidence of covalent bonding between the CuO NPs and MPC-SH. The XPS data are more consistent with the adsorption mechanism of attachment (Figure 2b) than the covalent bonding mechanism (Figure 2a). In the N 1s and S 2p regions, small peaks for nitrogen and sulfur appeared to be present in the survey spectrum (Appendix A), but these were not clearly detectable in the high-resolution spectrums for those regions. Since clear peaks would be expected from S and N atoms in PDA and MPC-SH, it is likely that the PDA and MPC-SH were hidden beneath one or more layers of CuO NPs. This is possible since the penetration depth for XPS is only ~5 nm [55], which is smaller than the size of the CuO NPs (~6–10 nm). The presence of PDA and MPC-SH on the membrane surface was ultimately confirmed by characterization methods, discussed later, which do have a deeper penetration depth.

The top surfaces of PES, PES/PDA, PES/PDA/MPC, and PES/PDA/MPC/CuO membranes were imaged by SEM at a magnification of 20,000× (Figure 6). Note that the magnification used was not high enough to resolve pores. Unfunctionalized and partially functionalized membranes had flat, featureless surfaces. However, for the PES/PDA/MPC/CuO sample, clusters of agglomerated CuO NPs were clearly visible and were dispersed fairly homogenously. Some clusters were as large as 500 nm, although most were on the order of ~100 nm, which is substantially larger than the 6–10 nm size of individual nanoparticles determined by TEM.

EDX results (Figure 7) showed that the surface of functionalized membranes contained substantial amounts of carbon (65.41 at%), nitrogen (8.33 at%), oxygen (24.53 at%), sulfur (0.32 at%), copper (0.91 at%), and platinum (0.5 at%). The mass of copper on the membrane was quantified by ICP-MS to be 72.5 ± 5.1 µg (see Section 3.4). The presence of copper provides further confirmation of the attachment of CuO NPs to the surface. Notably, there is a very high (26:1) nitrogen to sulfur ratio. Since PDA is the only component of the surface functionalization that contains nitrogen but not sulfur, this indicates the presence of significant amounts of PDA. Since the penetration depth for EDX (~2 μm) [56] is much greater than XPS (~5 nm), the PDA is detected with EDX, whereas it was not seen with XPS due to being obscured beneath the CuO NP layer. The small amount of sulfur (0.32 at%) is from the MPC-SH and/or the PES base layer. Also, a small amount of platinum is present from the sputter-coating process.

FTIR spectra were taken of raw MPC-SH and membrane samples at various stages of functionalization to confirm the attachment of MPC-SH to the membrane (Figure 8). MPC-SH showed a distinctive peak at 1730 cm^−1^, arising from the carbonyl group. This peak was not present in the spectrum of a plain PES membrane, but the peak was present (but small) in spectra of the PES/PDA/MPC sample and the fully functionalized PES/PDA/MPC/CuO sample. This peak confirms that the MPC-SH successfully attached to the PDA layer of the membranes and remained on the membranes after the addition of CuO NPs. Smaller peaks arise from C-H stretching (alkene) at 3030 cm^−1^ and C-H stretching (alkane) at 2853 cm^−1^ and are present in MPC-SH and membranes functionalized with MPC-SH. Since the alkene group in MPC reacts with 1,10-decanedithiol during thiolation, the peak at 3030 cm^−1^ is likely from a small amount of residual unreacted MPC.

Contact angle testing indicated a water contact angle of 97.1° ± 11.0 for a PES membrane, 69.6° ± 1.3 for a PES/PDA membrane, 68.1° ± 3.2 for a PES/PDA/MPC membrane, and 88.0° ± 0.6 for a PES/PDA/MPC/CuO membrane (Figure 9). This result shows that hydrophilicity increases with the deposition of PDA and MPC-SH. Some of the hydrophilicity is lost when the CuO NPs are added, although the membrane remains slightly hydrophilic overall. XPS data suggested that CuO NPs fully cover the surface of the membrane, blanketing the PDA and MPC-SH layers. Therefore, the CuO NPs appear to interfere with the hydration layer that would normally be formed around MPC-SH, resulting in a less hydrophilic membrane compared with the partially functionalized PES/PDA/MPC membrane.

### 3.3. Membrane Filtration

Dead-end filtration was performed for unfunctionalized PES and functionalized PES/PDA/MPC/CuO membranes (Figure 10). Functionalized membranes had a slightly lower pure water flux, 106.4 ± 15.3 L/(m^2^·h) (LMH), compared to unfunctionalized PES membranes, which had a flux of 110.1 ± 10.9 LMH. The sizeable standard deviations in flux for functionalized membranes could be largely due to the original variation in flux from membrane to membrane.

Surface functionalizations typically reduce pure water flux since there is increased hydraulic resistance from material added at the surface. For instance, Shahkaramipour et al. [29] coated PES UF membranes with a zwitterionic polymer and found that PWF declined by 36–62%, depending on coating parameters. For our PES/PDA/MPC/CuO membranes, the decline in PWF was comparatively low, at 3.4%.

During BSA filtration, BSA rejections were very high, at 99.29 ± 0.21% for the functionalized membranes and 99.45 ± 0.07% for the PES membranes. Functionalized membranes showed improved flux, at 48.4 ± 4.9 LMH (54.5% flux decline vs. PWF), compared to PES membranes at 35.6 ± 12.7 LMH (67.7% flux decline). For PES UF membranes, BSA fouling occurs mainly through cake formation and pore blockage [57]. The PDA/MPC/CuO functionalization reduced pore obstruction by increasing hydrophilicity. Similarly, Shahkaramipour et al. [29] found that during BSA filtration, zwitterion-modified membranes had less flux decline (38%) than PES membranes (53%).

After flushing the membranes with DI water in an inverted position, the recovery flux increased to 68.6 ± 4.4 LMH for functionalized (FRR of 64.5%) and 62.3 ± 9.8 LMH for unfunctionalized (FRR of 56.6%). After the hydrogen peroxide treatment, flux increased further to 73.8 ± 15.8 LMH (FRR of 69.3%) for functionalized membranes and 65.9 ± 10.6 LMH (FRR of 59.9%) for PES membranes. The improvement in flux after hydrogen peroxide treatment was not as large as Guha et al. [13], where flux returned at or above the original flux, although they did not test BSA as a foulant and used RO instead of UF membranes. Our FRR results are close to those of Krishnamurthy et al. [24], who mixed copper (I) oxide into PES UF membranes and measured FRRs of 64–77% after BSA filtration (depending on fabrication protocol).

In literature, values reported for FRR after BSA filtration and rinsing range extremely broadly for various types of unfunctionalized and functionalized PES UF membranes [23,58,59]. For unfunctionalized PES UF membranes, FRRs reported in a review article range from 15–80.6% [23]. Our data is above the average percentile range. For functionalized PES UF membranes, reported FRRs range from 39–48% on the low end, for PES membranes with amine-functionalized carbon nanotubes [58], to over 100% on the high end for PES membranes blended with novel triblock copolymers [59]. Even though it is not a direct comparison, our membranes functionalized with MPC-SH are 20% greater than the amine-functionalized and 40% lower than the PES triblock copolymer membranes.

### 3.4. Copper Leaching

Separate flux experiments were performed in which DI water was flowed through PES/PDA/CuO and PES/PDA/CuO/MPC membranes, and the effluent and post-flux membranes were analyzed for copper content with ICP-MS (Table 1). The copper content in the effluent was 7.4 ± 2.3 ppb for PES/PDA/CuO membranes and 11.8 ± 3.0 ppb for PES/PDA/CuO/MPC membranes. The calculated copper retention percentages were 98.0 ± 0.7% for PES/PDA/CuO membranes and 96.7 ± 0.5% for PES/PDA/MPC/CuO membranes. These results show that the CuO NPs are very stable on the surface of the membranes, regardless of whether or not MPC-SH was also present.

## 4. Conclusions

PES ultrafiltration membranes were functionalized with PDA, MPC-SH, and CuO NPs, then characterized and tested by dead-end filtration. These membranes could be used in ultrafiltration applications, such as water purification and protein separation. The main conclusions are as follows:1.Characterization by SEM/EDX, FTIR, and XPS confirmed the attachment of each component to the membranes. XPS indicated no signs of a thioether-copper bond, suggesting that the CuO NPs attached to the membranes by physisorption instead of covalent bonding.2.Separately, characterization of CuO/MPC nanoparticles by XPS did show a bond between the MPC-SH and copper. This indicates that it is possible for MPC-SH to attach to CuO NPs by the thiol group.3.Contact angle tests showed that PDA and MPC-SH improved the hydrophilicity of functionalized membranes from 97.1° to 68.1°. However, this improvement was somewhat reduced by the addition of CuO NPs, which increased the contact angle to 88.0°.4.Functionalized membranes had modestly improved performance during dead-end filtration with BSA (48.4 LMH), compared to plain PES membranes (35.6 LMH). After rinsing and cleaning with hydrogen peroxide, functionalized membranes had improved FRRs (69.3%) compared to plain PES (59.9%) membranes.5.Copper leaching was low for functionalized membranes (96.7% retained), indicating the stability of the CuO NP layer.

The above points suggest that membrane performance might be further improved by modifying the fabrication protocol so that CuO NPs are attached to the membranes before MPC-SH. This would enable the thiol groups of MPC-SH to covalently bond to the copper in the CuO NPs. In this approach, the MPC-SH would be fully exposed to the water layer, and therefore the hydrophilicity advantages of MPC-SH would be unhindered. Preliminary experiments have showed that a challenge of this approach is that the MPC-SH functionalization step destabilizes the PDA/CuO layer due to the basic environment of the reaction, so this would have to be overcome, perhaps by substituting another material for PDA.

## Figures and Tables

**Figure 1 membranes-12-00544-f001:**
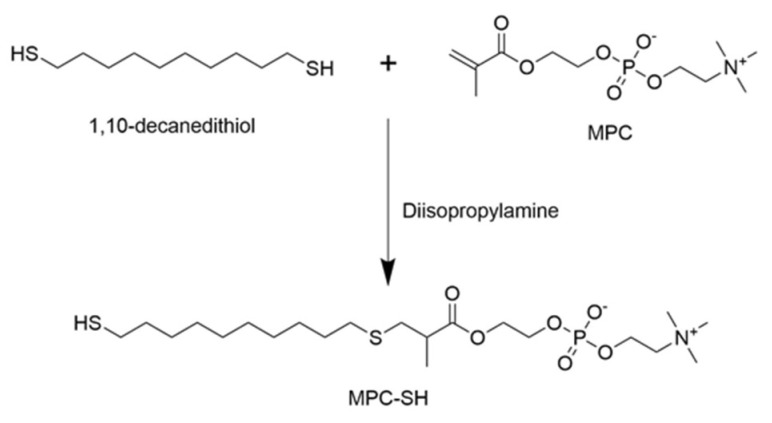
MPC-SH synthesis from 1,10-decanedithiol and MPC.

**Figure 2 membranes-12-00544-f002:**
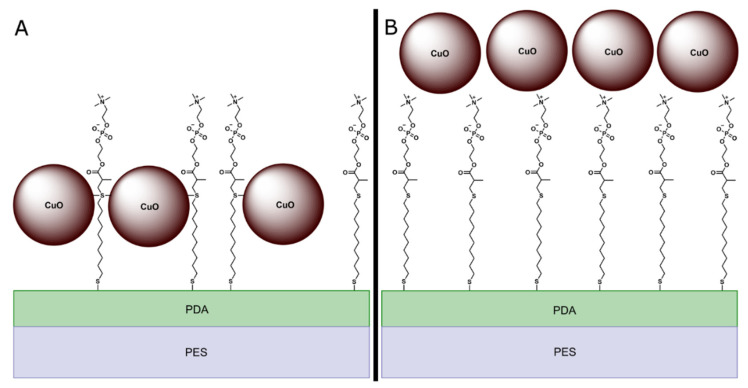
Possible structures of functionalized membrane top layer. In structure (**A**), CuO NPs are covalently bonded to the central sulfur atoms in MPC-SH. In structure (**B**), CuO NPs are adsorbed on top of the MPC-SH.

**Figure 3 membranes-12-00544-f003:**
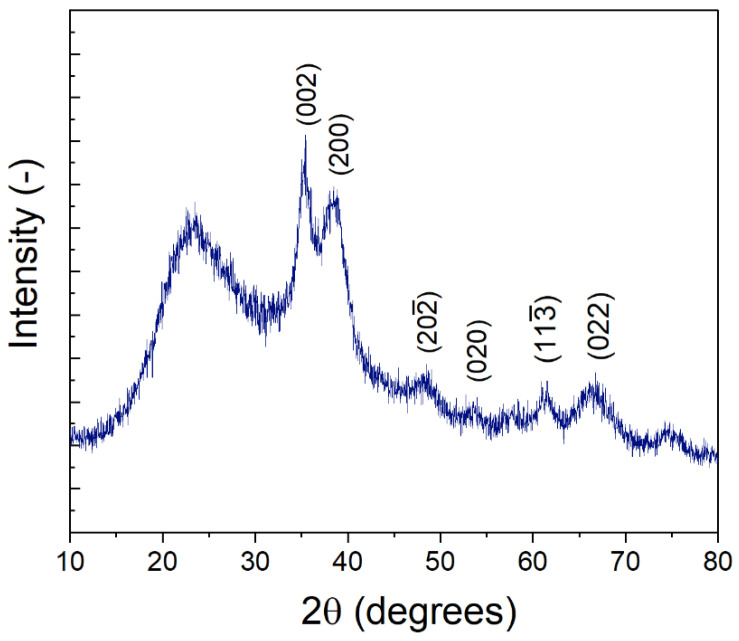
XRD spectra of CuO nanoparticles.

**Figure 4 membranes-12-00544-f004:**
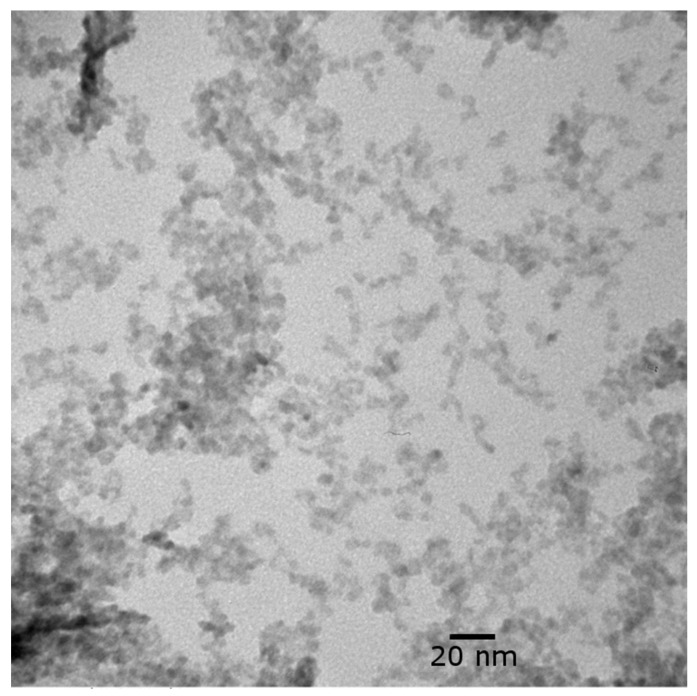
TEM image of CuO NP suspension at 500,000× magnification.

**Figure 5 membranes-12-00544-f005:**
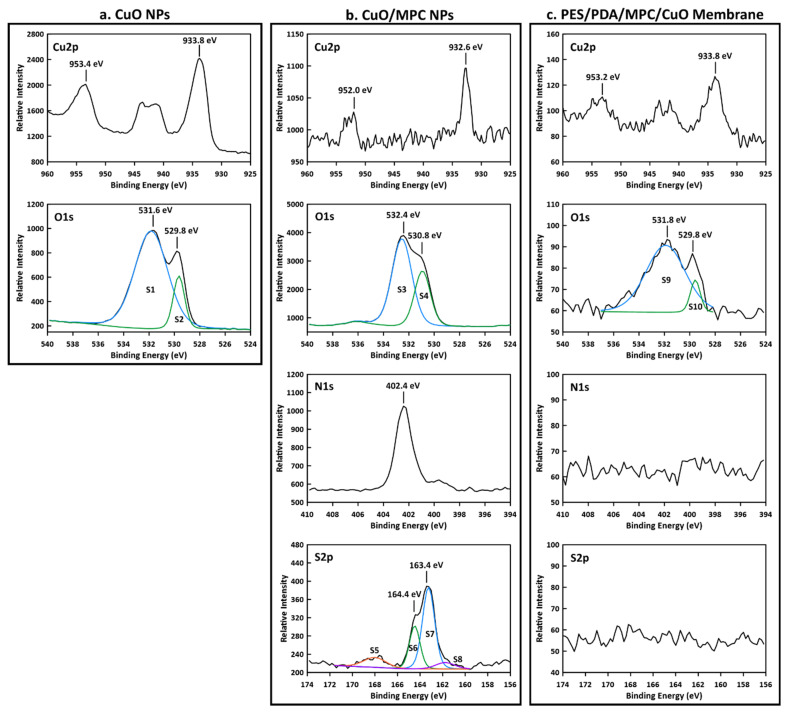
XPS spectra of nanoparticle and membrane samples. (**a**) CuO NPs spectra show peaks in the Cu 2p and O 1s regions consistent with copper (II) oxide. (**b**) CuO/MPC NPs spectra contain Cu2p_1/2_ and Cu2p_3/2_ peaks that are shifted to lower binding energy due to Cu-S bonds formed between the CuO NPs and MPC-SH. O 1s region peaks are from the C-O and C=O bonds in MPC-SH and an N 1s peak is from the alkyl ammonium group in MPC-SH. S 2p region peaks arise from oxidized sulfur (S5), C-S-C bonds (S6 and S7) and Cu-S bonds between CuO NPs and MPC-SH (S8). (**c**) PES/PDA/MPC/CuO membrane spectra contain peaks at virtually the same positions in the Cu 2p and O 1s regions as for CuO NPs and have no apparent peaks in the N 1s or S 2p regions, indicating that the CuO NPs cover the surface.

**Figure 6 membranes-12-00544-f006:**
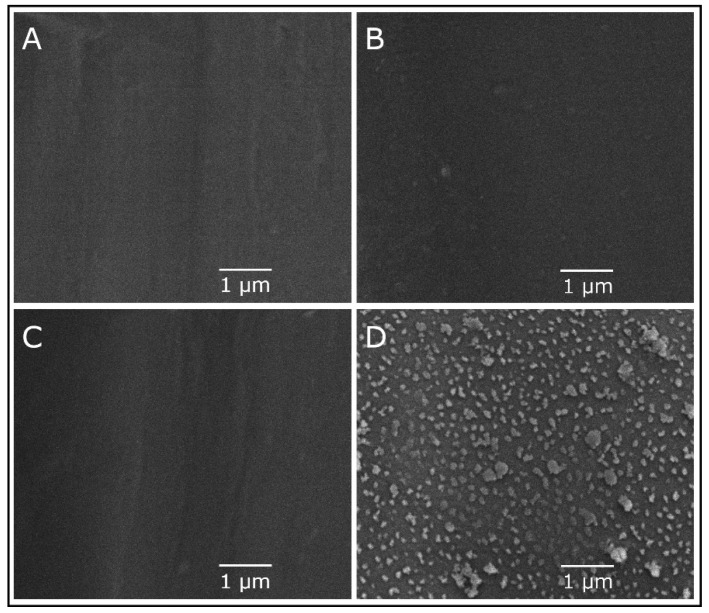
SEM image of the top surface of a PES membrane (**A**), PES/PDA membrane (**B**), PES/PDA/MPC membrane (**C**), and PES/PDA/MPC/CuO membrane (**D**) at 20,000× magnification.

**Figure 7 membranes-12-00544-f007:**
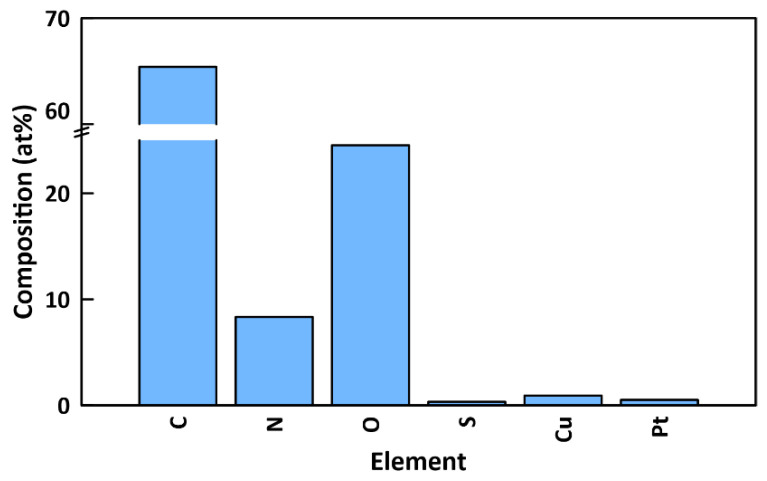
Elemental composition of top surface of functionalized PES/PDA/MPC/CuO membrane, as determined by EDX.

**Figure 8 membranes-12-00544-f008:**
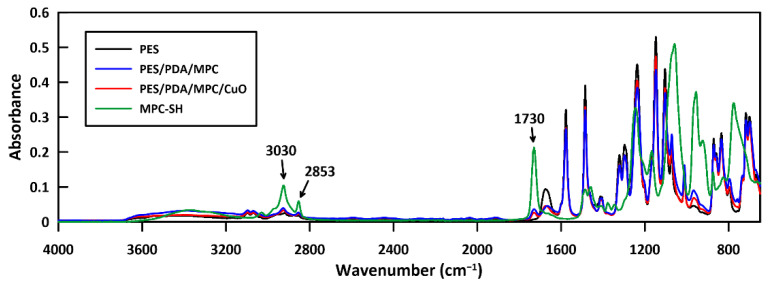
FTIR spectra of PES, PES/PDA/MPC, and PES/PDA/MPC/CuO membranes and pure MPC-SH. The peak visible at 1730 cm^−1^ is from the carbonyl group in MPC-SH. Peaks at 3030 cm^−1^ and 2853 cm^−1^ are C-H stretching peaks. These peaks occur in spectra of MPC-SH, PES/PDA/MPC, and PES/PDA/MPC/CuO, but not PES.

**Figure 9 membranes-12-00544-f009:**
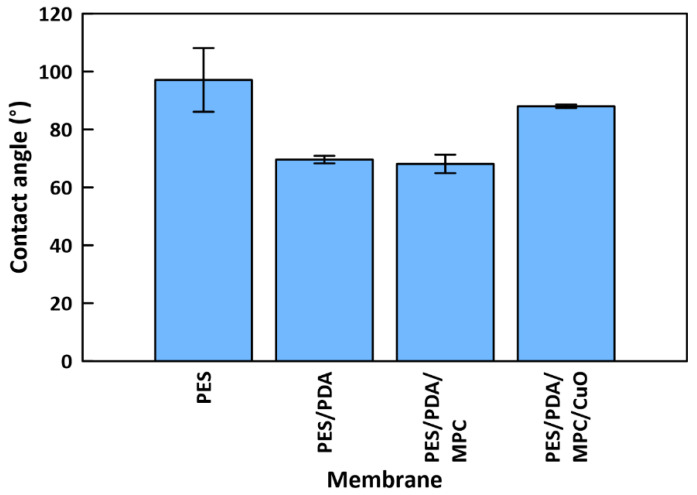
Water contact angles for membranes at various stages of functionalization.

**Figure 10 membranes-12-00544-f010:**
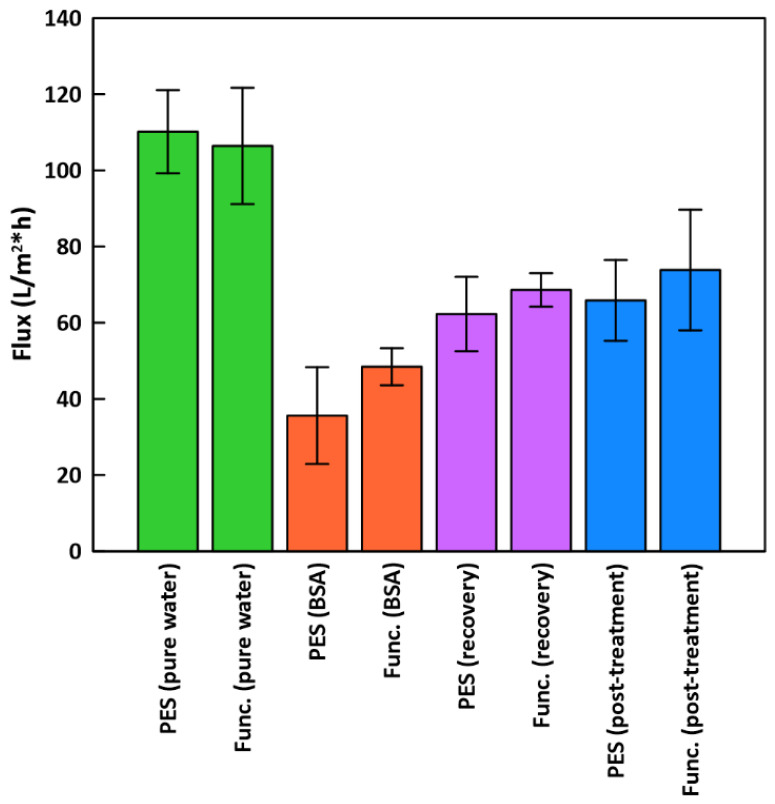
Dead-end filtration flux for PES and functionalized membranes. “Func.” results are for functionalized PES/PDA/MPC/CuO membranes.

**Table 1 membranes-12-00544-t001:** Concentrations and masses of copper in permeate and retained on membranes.

Membrane Type	PES/PDA/CuO	PES/PDA/CuO/MPC
Permeate Cu concentration (ppb)	7.4 ± 2.3	11.8 ± 3.0
Permeate Cu mass (µg)	1.5 ± 0.5	2.4 ± 0.6
Retained Cu mass (µg)	74.1 ± 8.1	70.2 ± 4.8
Starting Cu mass (µg)	75.6 ± 8.3	72.5 ± 5.1
Cu retention percentage	98.0 ± 0.7%	96.7 ± 0.5%

## Data Availability

The data presented in this study are available on request from the corresponding author.

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
