# Peer review of "Ultrafiltration Membranes Functionalized with Copper Oxide and Zwitterions for Fouling Resistance"

_membranes, 2022, doi:10.3390/membranes12050544_

Round 1

Reviewer 1 Report

This manuscript presents Ultrafiltration Membranes Functionalized with Copper Oxide and Zwitterions for Fouling Resistance. In this paper, the application of CuO as doped nanoparticles in ultrafiltration membranes has been analyzed. The authors should clearly explain the choice of CuO NPs and zwitterions for their experiments. References should be updated with more recent reports about novel methods and applications of CuO nanoparticles for reducing fouling and scaling. Please explain what is novel and what are the differences in this work in the introduction. The authors should clearly justify the choice of their methods and the advantages of their work.

The SEM or confocal microscopy surface images of the membranes with and without CuO NPs  before and after experiments should be provided for the information of shape and size distribution of the fouling and scaling and the change of the morphology of membranes after long time experiment. The preparing method and the optimization of CuO NPs and zwitterions ratio and processes should be presented in detail. English in some parts needs to be polished.

After major revisions and clear explanations, this manuscript can be published in the Membranes.

Reviewer 2 Report

The manuscript describes an ultrafiltration membranes functionalized with copper oxide and zwitterions for fouling resistance. Although the structure of modified membranes has been characterized, quantitative analysis data are obviously inadequate. Therefore, I do not recommend to publish it due mainly to several reasons as follow:

  1. The idea and results of this manuscript are well known. So, I cannot find new knowledge this study can provide to the reader.
  2. Although the structure of membranes has been characterized via various methods and demonstrated, almost any quantitative structural data cannot be provided. For example, what is the load of PDA and copper?
  3. The paper lacks systematic data to explain the influence of different modification steps and degrees on membrane properties.

Reviewer 3 Report

Detailed comments:
1. The English of the text should be checked

2. The novelty of manuscript should be highlighted more

3. At Introduction part the authors must be included more information about ultrafiltration in comparison
with other membrane processes (e.g. electrodialysis, reverse electrodialysis, nanofiltration,
microfiltration). Also, must be included more advantages and disadvantage of ultrafiltration in
comparison with other membrane processes, and more information about membranes that contains
nanoparticles. The following references can be included in the Introduction part to improve the quality
of manuscript,
because they provide relevant information:
✓ Polymeric membranes incorporated with metal/metal oxide nanoparticles: A comprehensive review.
Desalination 2013, 308, 1533.

✓
Study of Polyvinyl Alcohol-SiO2 Nanoparticles Polymeric Membrane in Wastewater Treatment
Containing Zinc Ions, Polymers, 2021, 13(11), 1875

✓ Removal of Cu2+, Cd2+ and Ni2+ ions from aqueous solution using a novel chitosan/polyvinyl alcohol
adsorptive membrane. Carbohydr. Polym. 2019, 210, 264273

✓ Biopolymeric Membrane Enriched with Chitosan and Silver for Metallic Ions Removal, Polymers
2020, 12, 1792

✓ Progress of Nanocomposite Membranes for Water Treatment. Membranes 2018, 8, 18.

✓ San copolymer membranes with ion exchangers for Cu (II) removal from synthetic wastewater by
electrodialysis, Journal of Environmental Sciences-China, 35, 2015, p. 27-37

✓ Electrodialysis Desalination for Water and Wastewater: A Review. Chem. Eng. J. 2020, 380, 122231.

4.
Lines 51, 55, 59, replace Guha and coworkers with Guha et al.; Arumugham and coworkers with
Arumugham et al., Krishnamurthy and coworkers with Krishnamurthy et al., please check in all
manuscript

5.
Line 135, what means “...at elevated temperature ...”? a value must be indicated. Also, for room
temperature must be indicated a value

6.
Lines 137-138, “The final nanoparticles were then transferred into the centrifuge tube and washed several
times using DI water” what means several times? a value must be indicated

7. Lines 143-144, “Samples were taken after 16 hours.” - the sentence seems unfinished, you need to
mention in what environment the membranes were kept and how long until they were characterized and
tested; membranes were taken for what?

8.
A scheme or a photo must be included at 2.6 Membrane Filtration
9.
Fig. 8 must be replaced with other more clarity, at a large scale. Also, include at axe, Wavenumber. The
figure shows significant changes, you must indicate other peaks, include comments on significant
differences between membranes

10. Line 424, at “In literature, values reported for FRR after BSA filtration...” the References must be
indicated

11. At Figure 10, on scale, at Flux replace LMH with L/m2*h

12.
More Conclusions with the best obtained results must be indicated
13. Comparison between the obtained results and measured in this study with other reported studies should
be done and included for more clarity (indicate values not just number of reference).

14. The possible other applications of the prepared membranes must be included

15. Same Reference are very old. The manuscript must contain the relevant information to be attractive for
readers (researchers),
because science has advanced, and the information indicated in the manuscript is
no longer valid. This part should include observed information, noted in the last 10-12 years.

Thank you very much.

Reviewer 4 Report

This study focused on surface modification of ultrafiltration membranes with copper oxide and zwitterions to enhance membrane antifouling stability. Membrane fouling is one of the most urgent problems in membrane technology, that’s why this paper is very interesting for researchers and shows a new approach for membrane improvement. Authors used the wide range of analytical methods to study membrane structure and properties.

I would recommend this paper for publication after following revisions.

  1. How did the authors carry out the H2O2 membrane treatment?
  2. 10, L. 352-354. “Since the penetration depth for EDX (~2 μm) is much greater than XPS (~5 nm), the PDA is detected with EDX, whereas it was not detected with XPS due to being obscured beneath the CuO NP layer”. However, there were no information about the thickness of the PDA or PDA/MPC/CuO layers. This information must be added.
  3. At FTIR spectra authors interpreted ony peak at1730 cm-1 Other signals that are different for a certain spectrum must be deciphered.
  4. What was the pore size of the selective layer of unmodified PES and functionalized membranes?
  5. The authors studied the water contact angle using settle drop method. Was it suitable to use it if the membrane selective layer had open pores? Did the authors carry out the long-term measurement of selective layer contact angle? What were the  changes of contact angle with time?
  6. What was the total decline ratio of reference PES and functionalized membranes?

Round 2

Reviewer 1 Report

The authors adjusted well according to part of my comments and showed some new SEM images. This manuscript can be published now in the Membranes.

Reviewer 2 Report

The quality of manuscript has been improved, therefore, I agree to recommend to publish. 

Reviewer 3 Report

Agree

Reviewer 4 Report

The authors satisfactorily responed to all comments and the paper can be accepted to publication in Membranes.